# Qualification programmes for immigrant health professionals: A systematic review

**Sidra Khan-Gökkaya**  *, Sanna Higgen, Mike Mösko

Department of Medical Psychology, University Medical Center Hamburg-Eppendorf, Research Group on Migration and Psychosocial Health, Hamburg, Germany

* s.khan@uke.de

**Data Availability Statement:** All relevant data are within the manuscript and its Supporting Information files.

**Funding:** The study was funded by the European Social Fund. SKG and MM received the funding.

## Abstract

### Background

Immigrant health professionals are a particularly vulnerable group in a host country's labour market, as they face several barriers when re-entering their occupations. International studies indicate that early interventions can increase the employability of immigrants. Qualification programmes are one of these early interventions that can support the re-integration of these health professionals into the labour market. The purpose of this review is to identify international qualification programmes for immigrant health professionals, analyse their content and evaluate their effectiveness.

### Methods

Six international databases (PubMed, Web of Science, CINAHL, PsychInfo, EBSCO and ProQuest Social Sciences) were systematically searched. The search terms were identified using the PICOS-framework. The review was performed in accordance with the Preferred Reporting Items for Systematic Reviews and Meta-Analyses (PRISMA). Articles were screened independently by two authors and discussed. Studies included in the final synthesis were assessed with the Mixed Methods Appraisal Tool (MMAT) and Kirkpatrick's Training Evaluation Model.

### Results

Out of 10,371 findings, 31 articles were included in the final synthesis. The majority of them were addressed to international health care professionals and doctors. Two of them were addressed to refugee doctors. Three types of programme elements were identified: teaching, clinical practise and social support. The programmes' length ranged from 2 days to up to 2.5 years. Despite recommendations in its favour, pre- and post-programme support was scarce. Results also indicate a lack of transparency and quality in terms of evaluation. Effectiveness was mostly observed in the area of language improvement and an increase in self-confidence.

The funders did not play any role in the study design, data collection, decision to publish, or preparation of the manuscript.

**Competing interests:** The authors have declared that no competing interests exist.

## Conclusion

This review points out the lack of systematically evaluated qualification programmes for immigrant health professionals. Programme providers should focus on implementing programmes for all health professionals as well as for underrepresented groups, such as refugees. In order to generate best practises it is necessary to evaluate these programmes. This requires the development of appropriate instruments when working with immigrant population in the context of educational programmes.

## Introduction

Over the past few years, the number of immigrants and refugees has increased worldwide [1]. In 2017 the number of international migrants reached 258 million, up from 220 million in 2010 and 173 million in 2000 [2]. Among those immigrants and refugees are skilled health professionals. As the global health care workforce is facing a shortage [3], several host countries health care systems focus on employing foreign-trained health professionals. However, the (re-)integration of immigrant and refugee health professionals is connected with multiple barriers. Good knowledge of the host country's official language [4] and the technical workplace-related language [5] are the first and foremost requirements for starting the (re-) integration process. As the professional standards for working in health professions differ between countries [6], additional training is required [7]. Due to a lack of supporting structures [5], this requirement is not easy to fulfil. Another barrier for immigrant health professionals is the unfamiliarity with the host country's health care system, procedures and standards [8]. These barriers may lead to deskilling [9], loss of self-confidence [10] and high levels of frustration [11]. The experience of racial discrimination [10] and structural inequality [7] in the host country's environment also hinder the integration process. Refugees experience additional barriers, as their access to labour market may be restricted [12], depending on the host countries legal framework. Furthermore, they must often go through a difficult recognition process [10] and/-or they may not be able to provide official documents [6].

In order to address these barriers and prepare immigrants for work, qualification programmes are strongly recommended [12, 13]. However, there is a broad range of programmes and designs. Some programmes have focused on the exchange between local employees and international health professionals in order to increase reflection on workplace differences, which have resulted in better workplace adjustments [14]. Other programmes, like the project "Placing Refugee doctors in Medical Employment" (PRIME) [15], have focused on clinical practise. PRIME facilitated a supervised training post for 25 refugee doctors. After participation in the project, 15 participants gained a job and were able to work again. Due to the increasing numbers of refugees in the past few years [16], several host countries have also decided to implement programmes. One such recent programme is hosted through a collaboration between the World Health Organization (WHO) in Turkey and the Ministry of Health in Turkey [17]. Whereas the Ministry of Health passed a law that allowed Syrian health professionals to work in Turkey, the WHO implemented a 7-week adaption training programme to prepare them for practise [17]. However, the outcomes of this programme have not yet been evaluated. There are also more extensive programmes expanding their design and integrating their programmes into residency trainings [18, 19] and/or combining them with courses on language and intercultural skills [20, 21]. Depending on the content, participation in such

programmes has resulted in higher chances of passing national examinations [22–27] that are required in order to work. Nevertheless, reviews on the effectiveness of qualification programmes for international medical graduates and health professionals criticise the methodological quality of the performed evaluations [28–30]. Furthermore, the examples of the programmes above show that the content of the programmes is diverse. Thus, in order to help educational providers design, implement and evaluate their programmes, this review aims at systematically identifying and analysing the content and effectiveness of evidence-based international qualification programmes for the labour market integration of immigrants in all health professions.

## Methods

This review was performed in accordance with the Preferred Reporting Items for Systematic Reviews and Meta-Analyses (PRISMA, [31]; S4 Table).

### Search strategy

The search was conducted via six international and interdisciplinary electronic databases during August 2017 and updated in September 2019 to include studies published during/after August 2017. The databases were PubMed, Web of Science, CINAHL, PsychInfo, EBSCO and ProQuest Social Sciences. No time limit was set and studies in English and German were included. The search terms were identified using the PICOS-framework [32] and adapted to each database. In order to identify programmes in the context of health care the PICOS-criteria comparison was replaced with context [33]. For each of the PICOS-criteria *(P: immigrant health professionals, I: qualification programme, C: health care, O: evaluation, S: primary and secondary articles)* synonyms were collected and reviewed by the co-authors´ group. The synonyms were then built into a search string according to each of the databases' rules and requirements (S1 File). Search terms were double-checked with MeSh terms. If a MeSh term did not cover any of the synonyms, it was added separately to the string. Search protocols documented the used search string, the dates, the database syntax requirements and the total number of articles found in the databases.

### Study selection

**Title and abstract screening.** During the first stage of title and abstract screening the inclusion criteria (Table 1) were simplified.

The articles found in the databases were exported to a reference management system in order to remove all duplicates and then exported for screening.

At the first stage of screening, the articles had to fit into the population and intervention of interest. The first 200 abstracts were screened and crosschecked by the first and second authors, reaching an interrater reliability of K = 0.7. The first author screened all abstracts for inclusion and exclusion criteria, whereas the second author screened one quarter of all of the retrieved references.

For the full-text (second stage) screening, the following inclusion criteria were applied according to the PICOS-tool:

Population: Due to a lack of evidence-based programmes for refugees, the search was extended to qualification programmes for immigrants as well as international and overseas trained health professionals from all health professions. To be included, these groups must have had personal migration experiences. Descendants of immigrants were excluded. Henceforth the term immigrant will be used for the target population as it reflects this shared experience of personal migration of a variety of groups. The second part of population referred to

**Table 1. Screening criteria for studies (S1 Table).**

| First stage of screening | |
|---|---|
| Population | immigrant health professionals, refugee health professionals, international, foreign and overseas trained health professionals |
| Intervention | programmes that aimed to prepare the population for working in health professions |
| **Second stage of screening** | |
| Population | 1. immigrant health professionals, refugee health professionals, international, foreign and overseas trained health professionals <br> 2. every health care profession according to the international labour organisation [34] |
| Intervention | 1. programmes preparing the population for working in health professions <br> 2. occupational specific educational programmes <br> 3. programmes focusing on the recognition and licensing of the population <br> 4. health profession specific language courses <br> 5. intervention and sample must exceed two days and two participants Exclusion criteria: <br> 6. programmes for groups that are already working in their original occupations |
| Context | 1. labour market integration into health professions and health context <br> 2. primary, secondary or tertiary care <br> 3. contact to patients or with machines in health care |
| Outcome | 1. qualitative or quantitative evaluations <br> 2. transparency in terms of evaluation methods |
| Study Design | 1. studies with primary and secondary data <br> 2. studies carried out in a qualitative or quantitative mannerstudies in German and English <br> 3. Exclusion criteria: <br> 4. Commentaries, newspaper articles, and policy papers |

occupational groups and health professions. Besides generic synonyms like medical or professional personnel every health care profession according to the international labour organisation [34] was used as a term in order to ensure that no health care profession would be excluded.

Intervention: Intervention was defined as programmes that aimed to prepare the population to work in health professions. Occupation-specific educational programmes and programmes focusing on the recognition and licensing of the population were also included. Language courses were only included if they focused specifically on medical and health professional language. If the population group in the programme was already licensed or even working in their professions, the programmes were excluded because a successful labour market integration was presumed in these cases.

Context: The context of the intervention was labour market integration into health care and health context. Health care was defined in a very broad sense, not only including professions with contact to patients, but also those working with machines in primary, secondary or tertiary health care. Professions such as social workers or teachers were excluded.

Outcome: The intervention needed to be qualitatively or quantitatively evaluated. It was of crucial importance that evaluation methods were transparently described. Furthermore, the intervention and the sample had to exceed two days and two participants.

Study Design: Articles in German or English with primary and secondary data that were conducted in a qualitative or quantitative manner were considered for this review. Commentaries, newspaper articles, and policy papers were excluded.

Additionally, records identified from two reviews [28, 29] on qualification programmes for international medical graduates (IMG) were included into the screening process.

The full text screening was conducted independently by the first and second authors. Regular meetings between the authors were held to discuss differences. After the screening process, data from the studies were transferred into an extraction sheet and crosschecked by members of the research group. Data extraction related to several categories such as study design, information regarding the target group and the sample, information referring to the intervention, statistical analysis, evaluation methods, qualitative and quantitative results and key conclusions. The studies were assessed with the Mixed Methods Appraisal Tool (MMAT) [35] and Kirkpatrick's Training Evaluation Model [36]. Kirkpatrick's Model assesses the scope of the evaluation performed in the programmes on four levels (Level 1: Reaction, Level 2: Learning, Level 3: Behaviour, Level 4: Results). The MMAT assesses the overall methodological quality of the studies according to four quality criteria, depending on the study design. According to these four quality criteria studies can be ranked from 25% to a maximum of 100%. All articles were independently assessed by the first and second authors and critically discussed to ensure consensus. Two raters reached an interrater reliability of K = 0.8 for the MMAT and K = 1 for Kirkpatrick´s Training Evaluation Model. Throughout the screening and assessment process, regular meetings between the authors ensured critical reflection on possible disagreements and the reaching of a consensus.

## Results

### Study selection

Out of initially 8,507 findings, more than 7,000 had to be removed as they did not match the inclusion criteria for the first stage but often focused on the health of refugees and their treatment as patients. Through the update in September 2019, an additional 1,864 publications were found. 171 articles were included in the final full-text screening. 140 articles had to be excluded, mostly because they were not evaluated, did not offer any kind of programme or did not focus on health professionals. Eventually, 31 articles were included in the final analysis (Fig 1) and synthesised descriptively.

### Study characteristics

The general characteristics as well as information related to structural aspects of the studies, content and evaluation are summarised in Table 2. Almost all of these studies (n = 28) were conducted in an English-speaking country. Two were conducted in Israel and one in Germany. They were mostly published in the 2000s (n = 14) and 2010s (n = 14). The majority of the programmes were addressed to international health care professionals (n = 25). Four studies used the terms *migrant* or *immigrant health professionals*, whereas two studies explicitly addressed refugees. Most of the programmes were designed for doctors (n = 22). Six studies were designed for nurses, two were open to all health care professions, and one was designed for physiotherapists.

### Programme design

The programmes' length ranged from 2 days [53] to up to 2.5 years in cases of special forms of residency [19]. Programme designs can be divided into three categories. The first category refers to programmes combining teaching and clinical practise (n = 11 [15, 18–21, 41, 42, 46, 51, 52, 54]). The second category refers to programmes only offering teaching (n = 10 [22, 25–27, 39, 43, 44, 47, 50, 55]) or practise (n = 2 [40, 48]), whereas programmes in the third category (n = 8 [14, 23, 24, 37, 38, 45, 49, 53]) offered primarily elements of social support, such as mentorship [23, 37], peer support [45], reflection through exchange with local staff [14], case

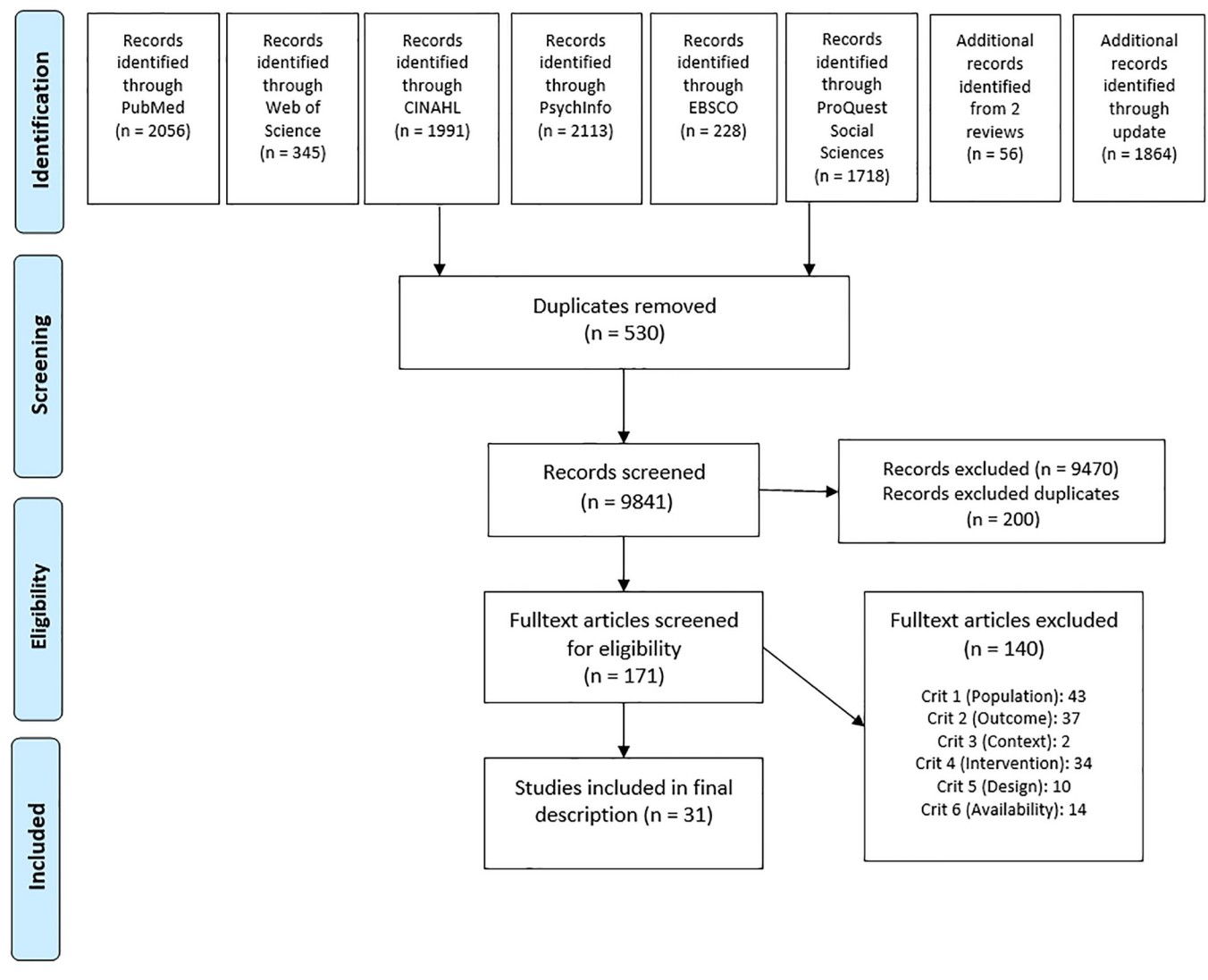

**Fig 1. PRISMA flowchart.**

management and counselling [38], career advice [45], social support or group activities. Within the third category, two programmes relied on media-based teaching only via videoconferences or web-tools (n = 2 [24, 49]). However, in most cases (n = 4 [23, 38, 45, 53]) these social support elements were combined with teaching.

## Teaching content

Most programmes focused on language, communication and consultation skills, including aspects of doctor-patient-relationship (n = 13 [18, 21, 23, 39, 41–43, 45, 47, 49–52], teamwork (n = 6 [15, 41, 42, 45, 47, 54]) and cultural and ethical aspects (n = 9 [18, 19, 21, 39, 41, 49, 50, 52, 53]). Medical standards, clinical practise (n = 10 [14, 22, 23, 27, 41, 42, 49, 52–54]) and the health care system (n = 8 [15, 20, 38, 41, 45, 50, 51, 53]) were also common topics. National examination preparation was likewise part of the delivered content (n = 10 [18, 20, 22–27, 50, 55]).

**Table 2. Study characteristics (S2 Table).**

| Reference | Host Country | Target Population | Study Design | Summary of Intervention | Sample | | Description of Evaluation methods | Summary of Outcomes |
|---|---|---|---|---|---|---|---|---|
| | | | | | total N | Age, Gender, Country of origin | | |
| Andrew, 2010 [18] | Canada/ Vancouver | International medical graduates (IMGs) | Non-randomised controlled trial | This programme was a family practise residency programme with a specific training site and teaching for IMGS in order to address more cultural, ethical, communication-related issues. | IMGs N = 371 Canadian: N = 313 | Age: M = 40 / Gender: no information / Country of origin: no information | In-training evaluation report (ITER) and results of Canadian Certification in Family Medicine (CCFP) examination pass rates between IMGs and control group | There were no significant differences in the In-training evaluation report between Canadian and IMG students. In passing the CCFP examination Canadians still were more successful (95%) than IMGs (58%). |
| Atack et al., 2012 [20] | Canada/ Ontario | Internationally educated nurses | Qualitative | This programme combined three elements: teaching, online based exercises and practise including English skills and an introduction to professional practise in Canada. | N = 62 (total) 1. Focus group sessions (N = 29) 2. Focus group (N = 19) 3. telephone interviews (N = 9) | Age: no information / Gender: 53 female, 9 male / Country of origin: no information | Focus groups and interviews at several points about individual feedback and programme strength and gaps | Programme enhanced participant's confidence; practise was seen as most valuable in adapting to host country's health care. |
| Daniel et al., 2016 [37] | Canada | Internationally educated health professionals (IEHPs) | Incidence Study | This programme introduced a Clinical Practise Facilitator (CPF) during an internship (including weekly classes) of IEHPs. The CPF had multiple roles: giving feedback to IEHPs, represent their interests, guide and encourage them. | N = 35 IEPs N = 37 CIs | Age: no information / Gender: no information / Country of origin: mostly Philippines (28%) and India (21%) | Self-developed questionnaire at the end of programme about the role of the CPF and benefits and challenges of the role | The versatile role of CPF was seen in several ways beneficial as for example to provide feedback, to answer questions and to support participants. Participants perceived cultural differences between the CPF and themselves as challenging. |
| Fernández-Peña, 2012 [38] | USA/ California | Immigrant health professionals | Incidence Study | This programme focused on case management, building up networks for migrant health professionals as well as introducing them to US health care practise and language through courses. | N = 10,476 | Age: 60% were between 30 and 49 / Gender: 72% female, 28% male / Country of origin: Mexico, Philippines, El Salvador, China, Peru, Colombia, Russia, India, Ukraine, Nicaragua, Iran, Haiti, Brazil, Guatemala | Demographic descriptive data (job post, exam taking rates, obtaining license, etc.) | WBI had a wide scope and reached a lot of immigrant health professionals. Approx. half of them succeeded in some way (validating their credentials, passed their exam, gained employment or higher positions). |
| Hawken, 2005 [21] | New Zealand/ Auckland, Wellington | Overseas-trained doctors (OTDs) | Non-randomised controlled trial | This programme combined teaching (consultation, communication, cultural issued and ethics) and supervised clinical practise. | N = 96 | Age: no information / Gender: 22 female, 74 male / Country of origin: Bangladesh, India, Sri Lanka, China, Egypt, Iraq, Iran, Singapore, Russia, Philippines, Serbia, Albania, Croatia | Pre-post and post course self-developed questionnaire for alumni's about the usefulness of the course, the participants' perception of their skills before and after the programme and suggestions for improvement | There was a significant increase (p<0.001) in participants comfort with their abilities to communicate effectively with patients in particular Maori patients (p<0.001). |
| Lujan & Little 2010 [25] | major city of the United-States | Migrated nurses | Mixed Methods | This programme was based on teaching with a focus on preparing for the state-approved examination. | N = 20 | Age: M = 28 / Gender: 19 female, 1 male / Country of origin: Mexico | Formative and summative evaluation through verbal short answers and written examination results | Half of the participants passed the NCLEX-RN test with a pass rate of 50% which is higher than the earlier reported pass rate of 22%. |

**Table 2.** (Continued)

| Reference | Host Country | Target Population | Study Design | Summary of Intervention | Sample | | Description of Evaluation methods | Summary of Outcomes |
|---|---|---|---|---|---|---|---|---|
| | | | | | total N | Age, Gender, Country of origin | | |
| Majum-dar et al. 1999 [39] | Canada/ Ontario, Toronto | Foreign medical graduates (FMGs) | Non-randomised controlled trial | This programme leaned on teaching through group sessions, simulated interviews and observation of videos focusing on communication and cultural aspects. | N = 24 (experimental group) N = 24 (control group) | Age: no information / Gender: 48% female, 52% male / Country of origin: mostly China, Vietnam, Egypt | Cross-cultural Adaptability Inventory (CCAI) was used to assess ones effectiveness in cross-cultural situations compared to a control group pre-test/post-test | Significant differences were found in two dimensions: emotional resilience (p<0.001) and perceptual acuity (p<0.03). |
| McGrat & Hender-son, 2009 [40] | Australia/ Queens-land | International medical graduates (IMGs) | Qualitative | This programme facilitated observerships and additional education with medical/ professional knowledge. | N = 9 | Age: range from 30–46 / Gender: 4 female, 5 male / Country of origin: mostly China (n = 6), Yugoslavia (n = 1), Philippines (n = 1) and Sri Lanka (n = 1) | Post course telephone-interviews about the participants' experiences with the programme, strengths and weaknesses of the programme | Programme was helpful and supportive for participants' entrance into workforce. Participants reported higher self-confidence, familiarity with the Australian health care and improvement of language and professional skills. |
| Ong & Paice, 2006 [15] | United Kingdom | Refugee doctors | Mixed Methods | This programme facilitated 'Senior House Officer' posts and introduced National Health Service (NHS) and other relevant issues through induction days. | N = 25 | Age: M = 41 years / Gender: 9 female, 16 male / Country of origin: mostly Iraq and Iran | Post course questionnaires and group discussions about participants view of the programme were evaluated along with job post rates | Participants reported improved confidence and knowledge. They were able to build networks. 15 of the 25 participants achieved substantive jobs within 12 months. |
| Parrone et al., 2008 [26] | USA/Midwest | Foreign nurses | Non-randomised controlled trial | This programme focused on preparing for the NCLEX-RN Examination and developing practical skills in a laboratory. Counselling and tutoring was provided when needed. | N = 67 | Age: range from 23–58 / Gender: 52 female, 15 male / Country of origin: mostly Philippines | Descriptive data about the correlation between attending the course, scoring rates of the HESI examinations and passing rates in the NCLEX-RN examination | There was a significant correlation (p<0.05) between HESI scores and NCLEX-RN pass rates. |
| Peters & Braeseke, 2016 [19] | Germany | Immigrant nurses | Mixed Methods | This programme consisted of theoretical (language, work, care) and practical training prior to a language and intercultural training at arrival. | N = 138 (interviews) N = 100 (questionnaires) | Age: no information / No information / Country of origin: Vietnam | Formative and summative evaluation through (group) Interviews and self-developed questionnaires about the participants' and facilities' experiences | The majority (92%) of the nurses completed the training and now work as nurses. Nurses were satisfied with the programme. Over 90% of the nurses approved the usefulness of intercultural training. |
| Sullivan et al., 2002, [41] | Australia/ New South Wales | Overseas-trained doctors (OTDs) | Non-randomised controlled trial | This programme combined teaching and supervised clinical attachment. | N = 66 | Age: female M = 37, male M = 36 / Gender: 58% female, 42% male / Country of origin: no information | Self-developed pre and post-test questionnaires, satisfaction sheets for daily sessions and a focus group at the end of the programme | Participants gained more confidence in their abilities to cope (p<0.004) and in relating with patients and peers (p<0,000), their communication (p<0.000) and their judgemental (p<0.046) skills. Participants had a greater understanding of the system and were less concerned about getting back to work. |

*(Continued)*

**Table 2.** (Continued)

| Reference | Host Country | Target Population | Study Design | Summary of Intervention | Sample | | Description of Evaluation methods | Summary of Outcomes |
|---|---|---|---|---|---|---|---|---|
| | | | | | total N | Age, Gender, Country of origin | | |
| Wright et al., 2011 [42] | Australia/ Gipps-land | International medical graduates (IMGs) | Mixed Methods | This programme offered simulated consultations along with meetings and web-based educational tools and a short period of observed practise. | N = 17 | Age: M = 35 | Self-developed questionnaires on meeting the learning objectives, pre post self- and external assessment through multisource feedback (MSF) and telephone interviews after the programme | Significant improvement was identified in three areas: technical skills, willingness and effectiveness when teaching colleagues and communication with carers and family. |
| | | | | | | Gender: 7 female, 10 male | | |
| | | | | | | Country of origin: Sri Lanka, the Philippines, Colombia, India, Bulgaria, Bangladesh, Iran, Afghanistan, Vietnam, China, Egypt and Bosnia | | |
| Baker & Robson, 2012 [43] | United Kingdom/ Scotland/ Dumfries and Galloway | International medical graduates (IMGs) | Mixed Methods | This programme focused on teaching language and consultation skills. | N = 14 | Age: no information | Pre-post language skills assessment and post course focus groups | There was a significant improvement in defining clinical problems (p<0.02) and explaining problems (p<0.004) to patients. 44% of the supervisors saw an improvement in language and consultation skills. |
| | | | | | | Gender: no information | | |
| | | | | | | Country of origin: India (n = 9), Pakistan (n = 2), Sri Lanka (n = 1), Libya (n = 1), Sudan (n = 1) | | |
| Bruce et al., 1974 [44] | United States/ Illinois | Foreign medical graduates (FMGs) | Non-randomised controlled trial | This programme was a language course designed for the needs of FMGs in speaking publicly. | N = 9 | Age: no information | Pre-post audio and video language assessment | The scores on audio and video performance before and after the programme showed significant improvement (p<0.005). |
| | | | | | | Gender: 5 female, 4 male | | |
| | | | | | | Country of origin: Korea (n = 5), Philippines (n = 1), Taiwan (n = 1), Egypt (n = 1), Iran (n = 1) | | |
| Cheung 2011 [45] | United Kingdom | Overseas-trained doctors (OTDs) | Mixed Methods | This programme combined teaching, peer support through other staff members and professional advice on career if needed. | N = 12 | Age: no information | Post course (self-developed) questionnaire, focus group and telephone interviews | Participants rated the course as relevant (M = 4.7 on a 5-point Likert scale), adequate (M = 4.2). Participants highlighted the peer support especially when the peers had the same cultural background as the participants. |
| | | | | | | Gender: no information | | |
| | | | | | | Country of origin: no information | | |
| Elis et al., 2005 [22] | Israel | Foreign graduate residents | Non-randomised controlled trial | This programme focused on teaching medical subspecialties and preparation for examinations. | Study group: N = 130 internal medicine residents; Control group: N = 405 residents | Age: range 28–53 | Self-developed feedback questionnaire post course, results in the Israeli examination compared to a control group | A high overall satisfaction score was given by the participants in response to the course (M = 4.28 on a 5-point Likert scale). Participants of the course had a significant higher chance of passing than the ones in the control group (41,7% vs. 30,4%; p<0.001). |
| | | | | | | Gender Study group: 74 female, 56 males | | |
| | | | | | | Country of origin: mostly Soviet Union | | |

(*Continued*)

**Table 2.** (Continued)

| Reference | Host Country | Target Population | Study Design | Summary of Intervention | Sample | | Description of Evaluation methods | Summary of Outcomes |
|---|---|---|---|---|---|---|---|---|
| | | | | | total N | Age, Gender, Country of origin | | |
| Gerrish & Griffith, 2004 [46] | United Kingdom | Overseas registered nurses | Qualitative | This programme combined three elements: an induction period, a supervised clinical practise and a mentorship by other nurses. Additional support was provided if needed. | N = 17 | Age: no information | Individual and focus group interviews at several times | Participants identified areas of success they connected to the programme which were most important to them (such as gaining professional registration, fitness for practise, getting employed and professional development in a valued organisational culture). |
| | | | | | | Gender: 17 female | | |
| | | | | | | Country of origin: China, Philippines, India, sub-Saharan Africa | | |
| Goldszmidt et al., 2007 [47] | Canada | International medical graduates (IMGs) and Internationally sponsored residents (ISRs) | Non-randomised controlled trial | This programme focused on English for medical purposes thus learning through clinical standardised patient scenarios. | ISRs N = 5, IMG N = 1 | Age: no information | Post programme feedback and pre-post self-evaluation of their skills | There was a significant increase in their communication skills ($p = 0.03$). |
| | | | | | | Gender: no information | | |
| | | | | | | Country of origin: no information | | |
| Greig et al., 2013 [23] | Canada | Internationally educated physiotherapists (IEPs) | Mixed Methods | This programme combined teaching (medical subjects and preparation for examinations) and a mentorship. | IEPs N = 124 | Age: no information | National exam results between control and intervention group | More than half of the participants (69/124) were integrated into workforce after the programme. Participation led to a 28% greater possibility of passing the written examination. |
| | | | | | | Gender: no information | | |
| | | | | | | Country of origin: UK (31%), India (21%), Australia (12%), Philippines (7%), US (5%), Brazil (5%), Iran (4%), Israel (3%), Netherlands (3%) | | |
| Harris & Delany, 2013 [14] | Australia/ Victoria | International medical graduates (IMGs) | Qualitative | This programme facilitated discussion and reflection sessions between IMGS and hospital staff. | No information | Age: no information | Feedback through evaluation cards after each session | Participants reported better adjustments to their new workplace and encouragement to critically reflect differences between their previous and current workplaces. |
| | | | | | | Gender: no information | | |
| | | | | | | Country of origin: no information | | |
| Horner, 2004 [48] | United Kingdom | Internationally recruited nurses | Non-randomised controlled trial | This programme facilitated a supervised practise programme. | IRNs N = 460 Mentors N = 100 | Age: no information | Self-developed post course questionnaire | Most of the participants that responded (response rate 23%) evaluated the programme as very beneficial and highlighted that having a mentor or some kind of support was important. Study days increased their confidence and knowledge. |
| | | | | | | Gender: no information | | |
| | | | | | | Country of origin: mostly from Philippines and Singapore | | |
| Lax et al., 2009 [49] | Canada/ Toronto | International medical graduates (IMGs) | Incidence Study | This programme consisted of a web-based e-learning programme focusing on communication and cultural issues through simulated doctor/patient scenarios, knowledge checks, reflective exercises and cases about medical topics. | S1: N = 20 S2: N = 42 S3: N = 33 | Age: no information | Usability test through a self-developed questionnaire and monitoring of participants' use of the web-based programme | Participants showed high levels of participation in the programme. Repeated participation and revision indicated knowledge building. |
| | | | | | | Gender: no information | | |
| | | | | | | Country of origin: no information | | |

(*Continued*)

**Table 2.** (Continued)

| Reference | Host Country | Target Population | Study Design | Summary of Intervention | Sample | | Description of Evaluation methods | Summary of Outcomes |
|---|---|---|---|---|---|---|---|---|
| | | | | | total N | Age, Gender, Country of origin | | |
| Ong et al., 2002 [50] | United Kingdom/ London | Overseas-trained doctors (OTDs) | Non-randomised controlled trial | This programme offered teaching courses on several topics such as communication, professional practise and health care system, multicultural issues and job searching skills. | N = 136 | Age: no information | Self-developed questionnaire after every daily session about the usefulness of the session | Topics were generally rated as useful (3.9–4.6. on a 5-point Likert scale). Most of the participants reported the programme was a useful introduction into NHS and workforce. |
| | | | | | | Gender: no information | | |
| | | | | | | Country of origin: mostly India and Nigeria | | |
| Ong & Gayen, 2003 [51] | United Kingdom/ London | Refugee doctors | Mixed Methods | This programme consisted primarily of clinical practise and was complemented by an induction day and an educational supervisor. | N = 29 | Age: mean male 32 / mean female 36 | Self-developed questionnaires at the end of the programme and analysis of discussions | All participants rated the scheme to be good or excellent (26/29). Most of the participants reported an increase in self-esteem and the feeling of belonging to a group. 17 of 29 doctors found a medical employment within 8 months. |
| | | | | | | Gender: 9 female, 20 male | | |
| | | | | | | Country of origin: Iraq (n = 14), Afghanistan (n = 5), Algeria (n = 2), Iran (n = 2), Uganda/ Congo/Russia/Libya/ Ethiopia (each n = 1) | | |
| Porter et al., 2008 [52] | United States, Omaha, Nebraska | International medical graduates (IMGs) | Mixed Methods | This programme alternated between theoretical approaches and clinical attachments. Furthermore it gave an orientation into residency and offered social support. | N = 11 (pre-post-test) N = 5 (interviews) | Age: no information | Medical knowledge and skills assessment through self-developed questionnaires pre and post course and interviews after the course | There was a significant increase in post-test scores for medical knowledge and skills such as discharge script writing and Subjective, Objective, Assessment, Plan (SOAP) note definition (p<0.05). Having a respectful and helpful instructor was emphasized by participants as well as their familiarisation with staff and health care. |
| | | | | | | Gender: 3 female, 8 male | | |
| | | | | | | Country of origin: mostly India | | |
| Romem & Benor, 1993 [27] | Israel | Immigrant doctors | Non-randomised controlled trial | This programme focused on courses on medical subjects through lecturing and problem oriented learning in small groups. Social group activities were integrated. | N = 273 | Age: 25–45 | Success rate in examination compared to a control group | The doctors who participated in the programme had a higher success rate at examination than that of the control group (p<0.019). |
| | | | | | | Gender: 142 female, 131 male | | |
| | | | | | | Country of origin: 226 from the Commonwealth Republics (82.8%), 32 Eastern European countries (11.7%), Rest: South America (5.1%) and one from Iran | | |
| Stenerson et al., 2009 [53] | Canada/ Saskatchewan | International medical graduates (IMGs) | Mixed Methods | This programme was based on an induction DVD and an orientation guide. Additionally a two day conference focused on clinical practise issues. | N = 107 | Age: no information | Post-course self-developed questionnaires and telephone interviews post course | Participants were satisfied with conference and 69% reported knowledge gains through conference and media based materials. These materials also supported in adjusting to the new workplace. |
| | | | | | | Gender: no information | | |
| | | | | | | Country of origin: no information | | |

(*Continued*)

**Table 2.** (Continued)

| Reference | Host Country | Target Population | Study Design | Summary of Intervention | Sample | | Description of Evaluation methods | Summary of Outcomes |
|---|---|---|---|---|---|---|---|---|
| | | | | | total N | Age, Gender, Country of origin | | |
| Watt et al., 2010 [54] | Canada/ Alberta, Calgary | International medical graduates (IMGs) | Non-randomised controlled trial | This programme combined a didactic course including role plays, case scenarios, practical exercises with a clinical placement including supervision and feedback. | S1: N = 39 S2: N = 235 | Age: S1: range 25–35 S2: M = 39 | Pre-post practicum ITER (S1) and pre-post English language assessment (S1 and S2). Post-course feedback by a self-developed questionnaire. Additionally there was a comparison group on Objective structured clinical examination (OSCE) data and language proficiency (S2) | There were significant changes in the language proficiency (p<0.001) pre and post-test. Improvements were also rated through ITER reports in clinical knowledge and skills (p<0.01). Participants of the programme outperformed other IMGS in their OSCE scores (they passed more OSCE station p<0.05 and had higher scores p.0.01). |
| | | | | | | Gender: S1: 25 female, 14 male; S2: 135 female, 100 male | | |
| | | | | | | Country of origin: S1: 17 countries (South American countries, Pakistan, China, Iran and African countries) S2: 22 countries of origin (primarily China, India, Pakistan, Iran, Eastern Europe and African countries) | | |
| Higgins et al., 2013 [24] | Australia/ Queensland | Specialist Int. medical graduates | Non-randomised controlled trial | This programme consisted of guided videoconferencing making exam topics a subject of discussion. | N = 166 | Age: no information | Participation and attendance of the media based programme modules associated with exam pass or fail rates | There was an association between tutorial participation and exam success. (Pass rate for those who participated 72%, for those who did not participate 41%). |
| | | | | | | Gender: no information | | |
| | | | | | | Country of origin: no information | | |
| Christie et al., 2011 [55] | Australia | International medical graduates (IMGs) | Mixed Methods | This programme consisted of a communication course focusing on language. | N = 8 | Age: no information | Anonymous post course questionnaires, assessment of language skills pre and post programme, focus group post course | There was improvement in pronunciation and non-verbal behaviour. Participants stated the training was useful. |
| | | | | | | Gender: no information | | |
| | | | | | | Country of origin: no information | | |

## Clinical practise

Clinical practise relates to any kind of clinical engagement–whether as an observer, an intern or as an employee. Clinical practise was supported by a mentor or a supervisor and emphasised as an important aspect of a programme. The role of the supervisor was emphasised by participants in one study [37] on the following terms: the supervisor should not only be a contact person to answer questions about clinical practise, but their role in the studies was also to give feedback, support, and promote participants´ skills and commitment. Results in one study explicitly reported on the lack of cultural competences of the supervisors, which resulted in the discouragement of the participants [37]. Two programmes solely offered clinical practise for three months [40, 48]. In terms of payment, one programme explicitly acknowledged the unpaid work of the participants during the clinical practise [40].

## Social support

Other elements of the programmes included peer support [45], the establishment of a network, especially with local staff [38], discussion and reflection with and between local staff [14], case management [38], counselling [19, 26, 41, 45], social support [52] and group activities [27].

## Study design and evaluation methods

Most studies either used a non-randomised controlled design (n = 6 [18, 22, 23, 27, 39, 54])
or a non-randomised one group design (n = 20 [15, 19, 21, 24, 26, 37, 38, 41–45, 47–53, 55]).
In terms of evaluation methods, 14 studies [18, 21, 24, 26, 27, 37–39, 44, 47–50, 54] used a
quantitative evaluation method, 13 used a mixed methods evaluation design [15, 19, 22, 23,
25, 41–43, 45, 51–53, 55] and four [14, 20, 40, 46] used a qualitative approach. Four studies
used validated instruments such as the Cross-Cultural Adaptability Inventory (CCAI) [39],
Objective structured clinical examination (OSCE) [54], In-Training Evaluation Report
(ITER) [18, 54] and Multisource feedback (MSF) [42]. These instruments asses one´s adapt-
ability to any culture (CCAI), communication and clinical skills (OSCE), overall performance
in care (ITER) and a 360-degree evaluation of the employee (MSF). Aside from these evalua-
tion methods, 15 studies used self-developed questionnaires, and ten studies used other kinds
of measurements (passing rates, web-based participation, video assessment, getting job posts,
etc.).

## Outcomes

The outcomes of the interventions can be divided into three categories: the improvement of
(i.) professional skills, (ii.) formal skills and (iii.) language skills. Within the first category of
improving professional skills (n = 20), participants reported on gaining knowledge about the
health care system and becoming familiarised with the system and the procedures. Studies also
indicated an increase in self-confidence amongst the participants and observed significant
improvements in terms of communication skills ($p<0.001$, [21]), emotional resilience
($p<0.001$, [39]) and perceptual acuity ($p<0.03$, [39]), coping with patients and peers
($p<0.000$, [41]), judgemental skills ($p<0.046$, [41]), defining and explaining clinical problems
($p<0.02$, [43]), script writing ($p<0.05$, [52]) and on In-training evaluation reports ($p<0.001$,
[54]).

The second category (n = 13) refers to formal resources, such as getting jobs, passing
national exams and establishing professional networks. Three studies proved higher chances
of passing the national examinations through their programmes [22, 26, 27], whereas one
study could not find any significant differences following programme completion [18].

The third category refers to outcomes only on the language skills level (n = 10). This
includes improvement in language, consultation and communication skills. Apart from the
significant changes in communication and writing skills that were reported in the first cate-
gory, one programme explicitly focused on audio and video performance of the participants.
They showed significant improvement in language skills ($p<0.005$, [44]), such as speaking, lis-
tening, comprehension and nonverbal communication.

## Quality assessment of the programmes

The majority of the studies (n = 17) evaluated on only one level of Kirkpatrick's training evalu-
ation model (Table 3): eight studies evaluated only on the level of reaction, 3 studies on the
level of learning, none on the level of behaviour and 5 on the level of results in terms of passing
rates of examinations or getting jobs. All the other studies (n = 14) evaluated outcomes on two
or more levels of Kirkpatrick's training evaluation model. The mean MMAT score (Table 3)
for qualitative (n = 4) and quantitative descriptive studies (n = 3) was 75%, for quantitative
randomised studies (N = 13) it was 50% and for mixed methods studies (n = 11) between 25%
(n = 7) and 50% (n = 6).

**Table 3. Quality assessment (S3 Table).**

| Qualitative | | | | Quantitative descriptive | | | |
|---|---|---|---|---|---|---|---|
| Reference | Kirkpatrick Level | MMAT Items* | Rating MMAT | Reference | Kirkpatrick Level | MMAT Items | Rating MMAT |
| Atack et al., 2012 [20] | 1 and 4 | 1.1. yes<br>1.2. yes<br>1.3. yes<br>1.4. can't tell | 75% | Daniel et al., 2016 [37] | 1 | 4.1. yes<br>4.2. yes<br>4.3. yes<br>4.4. no | 75% |
| McGrath & Henderson, 2009 [40] | 1 | 1.1. yes<br>1.2. yes<br>1.3. yes<br>1.4. yes | 100% | Fernández-Peña, 2012 [38] | 4 | 4.1. yes<br>4.2. yes<br>4.3. can't tell<br>4.4. yes | 75% |
| Gerrish & Griffith, 2004 [46] | 1 | 1.1. yes<br>1.2. yes<br>1.3. yes<br>1.4. no | 75% | Lax et al., 2009 [49] | 1 | 4.1. yes<br>4.2. no<br>4.3 yes<br>4.4 yes | 75% |
| Harris & Delany, 2013 [14] | 1 | 1.1., yes<br>1.2. yes<br>1.3. no<br>1.4. no | 50% | | | | |

*MMAT Items:
1. Sources of data relevant to objectives
2. Analysis process relevant to objectives
3. Consideration of findings relate to context
4. Consideration of findings relate to context

*MMAT Items:
1. Sampling strategy relevant to objectives
2. Sample representativeness
3. Measurements appropriate
4. Acceptable response rate

| Quantitative non randomised | | | | Mixed Methods | | | |
|---|---|---|---|---|---|---|---|
| Reference | Kirkpatrick Level | MMAT Items | Rating MMAT | Reference | Kirkpatrick Level | MMAT Items | Rating MMAT |
| Andrew, 2010 [18] | 3 and 4 | 3.1. no<br>3.2. yes<br>3.3. no<br>3.4. yes | 50% | Lujan & Little 2010 [25] | 4 | 1. 1 yes, 1.2. can't tell, 1.3. no, 1.4. no<br>4.1. yes, 4.2. yes, 4.3. yes, 4.4. yes<br>5. 1. yes, 5.2. yes, 5.3. no | 25% |
| Hawken, 2005 [21] | 1, 2, 3 | 3.1. yes<br>3.2. can't tell<br>3.3. can't tell<br>3.4. no | 25% | Ong & Paice, 2006 [15] | 1 and 4 | 1.1 yes, 1.2. yes, 1.3. no, 1.4. no<br>4.1., yes, 4.2. yes, 4.3. yes, 4.4. yes<br>5. yes, 5.2. yes, 5.5. no | 50% |
| Majumdar et al. 1999 [39] | 2 | 3.1. no<br>3.2. yes<br>3.3. yes<br>3.4. yes | 75% | Peters & Braeseke, 2016 [19] | 1 and 4 | 1.1.yes, 1.2. yes, 1.3. no, 1.4. no,<br>4.1.yes, 4.2. yes, 4.3. can't tell, 4.4. yes<br>5.1. yes, 5.2. yes, 5.3. no | 50% |
| Parrone et al., 2008 [26] | 4 | 3.1. can't tell<br>3.2. yes<br>3.3. no<br>3.4. yes | 50% | Wright et al., 2011 [42] | 1, 2, 3 | 1.1. yes, 1.2. yes, 1.3. yes, 1.4. yes<br>3.1. no, 3.2. yes, 3.3. no, 3.4. yes<br>5. 1 yes, 5.2. yes, 5.3. no | 50% |
| Sullivan et al., 2002, [41] | 2 | 3.1. no<br>3.2. yes<br>3.3. no<br>3.4. yes | 50% | Baker & Robson, 2012 [43] | 1 and 2 | 1. 1. yes, 1.2. yes, 1.3. yes, 1.4. yes<br>3.1. no, 3.2. no 3.3. no, 3.4. yes<br>5. yes, 5.2. yes, 5.3. yes | 25% |
| Bruce et al., 1974 [44] | 2 | 3.1. no<br>3.2. yes<br>3.3. no<br>3.4. yes | 50% | Cheung 2011 [45] | 1 | 1.1. yes, 1.2. can't tell, 1.3. no, 1.4. no<br>4.1. yes, 4.2. can't tell, 4.3. can't tell, 4.4. yes<br>5.1. yes, 5.2. yes, 5–3. no | 25% |
| Elis et al., 2005 [22] | 1 and 4 | 3.1. can't tell<br>3.2. yes<br>3.3. yes<br>3.4. yes | 75% | Greig et al., 2013 [23] | 1, 2, 4 | 1.1.yes, 1.2. can't tell, 1.3. can't tell, 1.4. no<br>3.1. can't tell, 3.2. yes, 3.3. yes, 3.4. yes<br>5. yes, 5.2. yes, 5.no | 25% |

(Continued)

**Table 3.** (Continued)

| | | | | | | | |
|---|---|---|---|---|---|---|---|
| Goldszmidt et al., 2007 [47] | 1 and 2 | 3.1. yes<br>3.2. no<br>3.3. no<br>3.4. yes | 50% | Ong & Gayen, 2003 [51] | 1 and 4 | 1.1 yes, 1.2. can't tell, 1.3. no, 1.4. no<br>3. 1 can't tell, 3.2. no, 3.3. can't tell, 3.4. yes<br>5.1. yes, 5.2. yes, 5.3. no | 25% |
| Horner, 2004 [48] | 1 | 3.1. yes<br>3.2. no<br>3.3. can't tell<br>3.4. no | 25% | Porter et al., 2008 [52] | 1, 2, 3 | 1.1. yes, 1.2. yes, 1.3. no, 1.4. can't tell<br>4.1. yes, 4.2. yes, 4.3. yes, 4.4. yes<br>5.1. yes, 5.2. yes, 5.2. no | 50% |
| Ong et al., 2002 [50] | 1 | 3.1. no<br>3.2. yes<br>3.3. can't tell<br>3.4. yes | 50% | Stenerson et al., 2009 [53] | 1 | 1.1. yes, 1.2. can't tell, 1.3. no, 1.4. no<br>3. 1 can't tell, 3.2. yes, 3.3. can't tell, 3.4. yes<br>5.1 yes, 5.2 yes, 5.3 no | 25% |
| Romem & Benor, 1993 [27] | 4 | 3.1 no<br>3.2. yes<br>3.3. no<br>3.4. yes | 50% | Christie et al., 2011 [55] | 1 and 2 | 1.1. yes, 1.2. no, 1.3. no, 1.4. no<br>3. 1 can't tell, 3.2. yes, 3.3. can't tell, 3.4. yes<br>5.1. yes, 5.2. yes, 5.3. no | 25% |
| Watt et al., 2010 [54] | 1 and 2 | S1: 3.1. can't tell<br>3.2. yes<br>3.3. can't tell<br>3.4. yes<br>S2: 3.1. can't tell<br>3.2. yes<br>3.3 can't tell<br>3.4. yes | 50%<br>50% | | | | |
| Higgins et al., 2013 [24] | 4 | 3.1. can't tell<br>3.2. yes<br>3.3. can't tell<br>3.4. yes | 50% | | | | |

| | |
|---|---|
| *MMAT Items:<br>1. Low-biased way of recruiting<br>2. Measurements appropriate<br>3. Consideration of differences between groups<br>4. Complete outcome data | *MMAT Items:<br>1. Mixed methods research design relevant to objectives<br>2. Integration of results relevant to objectives<br>3. Consideration of limitations associated with this integration |

## Discussion

This review aimed to identify evidence-based qualification programmes for immigrant health professionals and analyse their effectiveness. Previous research on the effectiveness of labour market programmes for all immigrants in Europe suggests that only wage subsidies positively influence the unemployment of immigrants [56]. However, as highly skilled professionals tend to remain in jobs which they are overqualified for, the question of how to successfully support their re-integration into labour markets arises. Research on IMGs´ transition indicates that qualification programmes surely play a role in the adjustment of IMGs and state that ongoing support is crucial for the success of such [28]. Nevertheless, research to date was unable to determine the effectiveness of programmes, as they lacked systematic evaluations. Hence, this review focused only on evidence-based programmes that transparently named evaluation methods. However, after becoming familiarised with the studies and assessing their quality, it became apparent that the risk of bias in the included studies was high and/or in many cases not sufficiently reflected upon. Additionally, due to a lack of reporting in the included studies, there may be a risk of incomplete or missing data in this review especially referring to the programmes design and content. In the context of programme design and content, it is important

to reflect on the social context of the programmes. National examinations, licensing procedures and other requirements may influence the purpose of the programmes and correspond to national requirements. However, in this study no country-specific patterns could be identified. Therefore, results on programme design, content and effectiveness can help educational providers design, implement and evaluate their programmes so that several aspects may be replicated in further studies.

## Programme design

Only one programme [38] in this review explicitly offered advice to participants about their career strategy before starting the qualification programme, although providing assistance for participants prior to the programme is recommended [12]. This may not only be helpful in terms of establishing individual career plans [12] but also in reducing barriers for participation, such as financial issues [38]. Regarding the core elements of the programmes, three components were identified: Teaching, clinical practise and elements of social support. These elements were either provided in combination or separately depending on the intervention aim. As language competencies are the first requirement for a successful labour market integration, it is not surprising that language and communication–including aspects of doctor-patient relationship, cultural issues and teamwork–seem to be the most important topic in the curriculum. However, it is surprising that only ten programmes aimed to prepare participants for examinations, even though passing national exams is a formal requirement on the path of labour market integration for health professionals [13]. This rare focus on exam preparations may be explained by the fact that a certain language proficiency is required in order to pass the exams, which is why programme providers focus primarily on language skills. As mentioned above, studies in this review mostly reported successful outcomes. When it comes to clinical practise, one study reported challenges between participants and supervisors who lacked cultural competencies [37], whereas in another study participants emphasized support from peers of the same cultural background as being helpful [45]. This underlines the role of local employees and health care providers who can function as facilitators. They can contribute to the success of labour market integration through a cultural competent attitude that supports the integration of immigrant health professionals [28]. It is also consistent with the claim that organisations need to promote an interculturally aware and sensitive atmosphere in order to give immigrant health professionals a sense of being accepted [28]. Furthermore, local supervisors, mentors or buddies can become trustworthy go-to persons in situations of doubt and provide the opportunity to try out tasks in a safe environment [28]. In addition, it can be assumed that they serve as the initial network in the clinical environment that may influence the target populations' career in terms of long-term sustainability. One limitation about the programmes was that it remained unclear whether there was any support for participants following successful completion of the programme and whether long-term networks were established via these programmes that could increase the cultural and social capital of participants and contribute to the outcomes identified in the second category of formal resources.

The appropriate length of a programme could not be determined due to a lack of reporting in the studies. Although there is no evidence on how long it takes health professionals to adapt to their new environment [57], results indicate that most providers prefer a programme of three to four months in duration. This duration is in line with recommendations given by the European Union for the labour market integration of refugees, stating that programmes with a duration of more than one year delay the transition to employment [12].

In general, the concepts of all reviewed programmes revealed a deficit-oriented view based on the assumption that immigrants come from countries with differing standards that need to

be adapted to those of the host country by means of these programmes. However, at the same time these professionals bring competencies and work experience, which are often not valued in the host country [11]. Similarly, previously gained competencies were not made visible in the programmes and thus not explicitly acknowledged. But with regard to the reported stress factors such as deskilling and high levels of frustration, as well as with regard to the outcomes that reported an increase in self-confidence, it appears that more positive affirmation and visible empowerment is needed in order to positively influence labour market integration [58]. Offering social support contributes to addressing this need. Nonetheless, to go even further, programme providers and organisations are responsible for creating an appreciative and empowering working and learning environment [28] in order to prevent immigrant health professionals from feeling like second-rate employees [59].

## Programme effectiveness

Approximately a quarter of the over 170 studies had to be excluded from this review in the second screening phase due to a lack of transparency in terms of evaluation methods. Based on the included studies, a general trend was observed in three different outcome dimensions: the improvement of (i.) professional skills (ii.) language skills and (iii.) the acquisition of formal qualifications. Although the sorting of the outcomes into these three dimensions should be interpreted with caution, as they are intertwined, they had an increase of self-confidence among the participants and their familiarisation with the health care system in common. As the loss of self-confidence and deskilling are reported stress factors for immigrants, it can be assumed that such programmes are at least helpful in counteracting these stress factors. However, in what way they contribute to a long-term successful labour market integration and how well immigrant health professionals adjust to their new working environment cannot be determined with certainty. Although a certain lack of evaluation methods and significant outcomes is consistent with previous research in this field [28–30], it raises the question about the appropriateness of the existing evaluation methods for the target group, as the instruments used to date in this field of research have limited or untested validity and reliability [60], and self-developed questionnaires are unreliable [61]. This may be one explanation for the poor quality of the studies assessed. Another reason for the poor assessment is the appropriateness of the MMAT tool in this context. If missing information from the studies was not traceable, studies received lower scoring rates due to information resources but not necessarily due to a poor methodological quality. Another challenge in applying this tool was selection bias for quantitative non-randomised studies. When working with immigrant health professionals, providers may not always be able to randomly choose participants. So during quality assessment, we were generally unable to definitively answer the question referring to selection bias, thus certain studies were rated poorly. Also, in most of the programmes there was no control group, which always led to one question (MMAT Item 3) remaining unanswered, thus resulting in assessment indicating poor methodological quality. The same applies to mixed method studies, where generally question three on appropriateness of reflection upon triangulation methods could not be answered as the term "appropriate" in its item is not clearly defined. Nevertheless, through quality assessment it can be concluded that there is a lack of systematically evaluated programmes without a high risk of bias. One possible explanation for this may be that programme providers' primary focus is not to conduct a scientific research but to promote the hands-on re-integration of immigrant health professionals. Despite this, in this review studies were only included if they were evaluated and published. Due to a lack of resources, we excluded grey literature. However, a number of qualification programmes are delivered by governments or non-governmental organisations who do not publish in scientific journals.

Thus, it should be noted that more programmes for the labour market integration exist that are helpful in some ways but have not been evaluated or published yet.

Despite the above listed challenges, a broad range of programmes was able to be identified. At the same time major blind spots in the field of qualification programmes became apparent. Out of the 31 included programmes, two were addressed to refugees–more precisely they were addressed to refugee doctors as part of a larger National Health Service (NHS) initiative aiming at getting refugee doctors back to work. This reveals a threefold gap in this field of research: (1) a lack of programmes for refugee health professionals (2) a lack of programmes for all health professionals (3) a lack of programmes that are systematically evaluated. The first gap refers to a general lack of programmes for refugees. This may be due to the fact that the latest included programme in this review dates back to 2016. The numbers of refugees increased between 2012 and 2015 [16] and raised humanitarian issues prior to issues of labour market integration [62]. Nevertheless, considering their labour market integration, there is evidence that refugees are confronted with more barriers than immigrants, due to their sudden flight and legal restrictions [6, 13]. These barriers may particularly affect refugee women, as they have poorer labour market outcomes [12]. Therefore, programme providers should consider the specific barriers for women and refugees in order to ensure an equitable access to labour market [7, 63]. The second gap refers to a lack of programmes that are supportive to all health professionals, although there is evidence that transition needs of doctors and nurses are similar and that exchange between professions is fruitful in terms of acculturation [28]. As skilled labour shortage does not only apply to doctors but also to other professions [3], programme providers should consider partly opening up programmes to involve all health care professions, instead of focusing on doctors. The third gap refers to a lack of programmes that are systematically evaluated. Consequently, the development of appropriate instruments for working with immigrant population in the context of qualification programmes should be promoted by future researchers.

## Strengths and limitations

The major strength of this review is the focus on a large group–not only international medical graduates but also immigrants and refugees and the consideration of their special needs. Furthermore, international programmes for all health professionals were included and interdisciplinary databases were used to consider programmes from all fields. Since there was no time limit set and due to the use of broad search terms, we were able to systematically analyse the content and the outcomes of the programmes. The analysis was also supported through quality assessment and the continuous reflection between the co-authors in order to ensure high quality of the findings. Nevertheless, there is a certain risk of bias in this review in terms of the population. Due to a lack of programmes for refugees, the search was extended to immigrant and international health professionals, although due to their flight and the circumstances of their flight, refugees may face even more or different challenges than international health professionals [6]. Another limitation of this review is that only studies in German and English were included in the analysis, and studies published in other languages are missing. Correspondingly, in this review there is only a representation of programmes conducted in the Global North, despite the fact that ten of the twenty largest destination countries for migrants worldwide are located in countries of the Global South [2].

## Conclusion

This study summarises evidence-based qualification programmes for immigrant health professionals and analyses their content and outcomes. Courses on communication, medical

standards and cultural aspects were frequently offered. Depending on the aim of the intervention they were combined with clinical practise or elements of social support. Effectiveness was mostly observed in the area of language improvement and in an increase of self-confidence. Nevertheless, the quality assessment of the studies pointed out a lack of transparency in terms of evaluation methods. Results also indicate a lack of evaluated programmes for all health professionals and refugees. Thus, educational providers should focus on implementing cross-occupational programmes, considering the special needs of subgroups, such as refugees, and evaluate their programmes in order to generate best practises.

## Supporting information

**S1 File. Search strings.**
(DOCX)

**S1 Table. Screening criteria.**
(DOCX)

**S2 Table. Study characteristics.**
(DOCX)

**S3 Table. Quality assessment.**
(DOCX)

**S4 Table. PRISMA Checklist.**
(DOC)

## Author Contributions

**Conceptualization:** Sidra Khan-Gökkaya, Mike Mösko.

**Data curation:** Sidra Khan-Gökkaya.

**Formal analysis:** Sidra Khan-Gökkaya, Sanna Higgen.

**Investigation:** Sidra Khan-Gökkaya.

**Methodology:** Sidra Khan-Gökkaya, Sanna Higgen, Mike Mösko.

**Supervision:** Mike Mösko.

**Visualization:** Sidra Khan-Gökkaya.

**Writing – original draft:** Sidra Khan-Gökkaya.

**Writing – review & editing:** Sidra Khan-Gökkaya, Sanna Higgen, Mike Mösko.

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
