## [Decision Letter · Decision Letter 0]

19 Sep 2019

PONE-D-19-22024

Qualification programmes for immigrant health professionals: a systematic review

PLOS ONE

Dear Sidra Khan-Gokkaya,

Thank you for submitting your manuscript to PLOS ONE. After careful consideration, we feel that it has merit but does not fully meet PLOS ONE’s publication criteria as it currently stands. Therefore, we invite you to submit a revised version of the manuscript that addresses the points raised during the review process.

We would appreciate receiving your revised manuscript by 20th October. To enhance the reproducibility of your results, we recommend that if applicable you deposit your laboratory protocols in protocols.io, where a protocol can be assigned its own identifier (DOI) such that it can be cited independently in the future. For instructions see: http://journals.plos.org/plosone/s/submission-guidelines#loc-laboratory-protocols

We look forward to receiving your revised manuscript.

Kind regards,

Sharon Mary Brownie

Academic Editor

PLOS ONE

Additional Editor Comments:

The author/s are requested to pay careful attention to reviewer comments. Please re-submit with a detailed response to each comment including comprehensive response to the reviewer request that your literature search is updated to include work published 2017-2019. Also consider seeking specialist assistance in respect to how the review of studies are presented in the manuscript. Specialist English review and edit is also recommended.

2. Please remove your figure 1 from within your manuscript file, leaving only an individual TIFF/EPS image file, uploaded separately as a figure file.  This will be automatically included in the reviewers’ PDF.

3. Please ensure that you refer to Figure 1 in your text as, if accepted, production will need this reference to link the reader to the figure.

Reviewers' comments:

Reviewer's Responses to Questions

**Comments to the Author**

1. Is the manuscript technically sound, and do the data support the conclusions?

Reviewer #1: Yes

Reviewer #2: Yes

2. Has the statistical analysis been performed appropriately and rigorously? 

Reviewer #1: No

Reviewer #2: Yes

3. Have the authors made all data underlying the findings in their manuscript fully available?

Reviewer #1: Yes

Reviewer #2: Yes

4. Is the manuscript presented in an intelligible fashion and written in standard English?

Reviewer #1: Yes

Reviewer #2: Yes

5. Review Comments to the Author

Reviewer #1: This was an interesting review and I enjoyed reading your manuscript. However, the following comments need to be addressed:

1. The search needs to be updated as the last search was conducted "during August 2017" (page 5, last paragraph). The search is dated and it will be interesting to see the articles published between August 2017-August 2019.

2. In the methods section (page 5-8), there was no mention of an analysis plan. Did the authors employ a narrative synthesis?

3. Additionally, how were the quality of the included articles appraised with respect to bias? Was there any quality assessment tool/instrument used, for example Newcastle-Ottawa Quality Assessment Scale? If so, the authors need to report this and also add a column to indicate same in Table 2.

4. The authors have highlighted using the Kirkpatrick's training evaluation model and created a heading 'quality assessment' page 23 (line 271). I will recommend the heading to be rephrased along the lines of 'quality assessment of training reported in the included studies' or 'evaluation of training in the included studies' (these are merely suggestions). The authors can come-up with a more suitable heading. But traditionally, quality assessment in systematic reviews are usually alluded to assessing the methodological rigor, bias, and reporting of the studies included in the review.

5. In Table 2, pages (11-20), it will be useful to have the First author's surname and year of publication in the first column and then add the reference next to it. For example, Andrew, 2010 [18]. This will be more informative for the reader as opposed to the current form.

6. A column for study design needs to be added in Table 2 (pages 11-20).

7. The 'quality assessment' results describing the Kirkpatrick's training evaluation model (pages 23-25) will be difficult to comprehend by a lay reader, the authors need to explain to the readers what the %s or ratings mean and their implications. Additionally, the references should be substituted with the First Author's surname and year of publication so the reader knows the score for each included study.

8. The manuscript should be proof-read once again for punctuation and grammatical issues.

8. The document

Reviewer #2: Thank you for requesting me to review this manuscript. The study, a systematic review of Qualification programs for immigrant health professionals addresses a very important area given the number of increasing refugees and immigrants globally. The study analyzed evidence based qualification programs for all health professionals. The selection process of the articles is very clear, as well as the assessment of the quality. The findings are summarized well. The use of an educational evaluation framework together with the MMAT strengthened the review.

It was interesting to note that only 2 articles addressed refugees given the current number of refugees. The introduction notes the number of international migrants, perhaps the number of refugees could also be noted, unless they are included in the migrants category.

The discussion section on page 29 line 378 includes aspects of limitations. Perhaps these could be moved to pages 30 -31 which discusses the strengths and limitations of the study.

There are minor edits that were noted, like the mean age of females in 51 on page 17. On page 22, revisit the abbreviations used to make sure they match content under discussion.

6. PLOS authors have the option to publish the peer review history of their article (what does this mean?). If published, this will include your full peer review and any attached files.

Reviewer #1: No

Reviewer #2: No

---

## [Author Response · Author response to Decision Letter 0]

21 Oct 2019

Dear Sharon Mary Brownie, Dear Reviewers, 

On behalf of Sanna Higgen and Mike Mösko I would like to express my sincerest thanks for your valuable support, your advice and your comments which we carefully considered within the process of our revision. 

I take this opportunity to address the points raised by the academic editors and reviewers.

Reviewer 1. P1: The search needs to be updated as the last search was conducted "during August 2017" (page 5, last paragraph).

Reply: We are very grateful for this important advice. We have now updated the literature search including work published between August 2017 and August 2019. We have uploaded the updated PRISMA flowchart (Fig 1) and the updated supporting information (S1 Search strings). 

Reviewer 1. P2: In the methods section (page 5-8), there was no mention of an analysis plan. Did the authors employ a narrative synthesis?

Reply: We have added a sentence (page 10) on the analysis plan. 

Reviewer 1. P3: Additionally, how were the quality of the included articles appraised with respect to bias? Was there any quality assessment tool/instrument used, for example Newcastle-Ottawa Qual-ity Assessment Scale? If so, the authors need to report this and also add a column to indicate same in Table 2.

Reply: We agree that it is very important to assess the quality of the studies with respect to bias. In our study, the Mixed-Methods Appraisal Tool (MMAT) was used to assess the quality of the studies. The MMAT is to date the only tool that can be applied to qualitative, quantitative and mixed-methods studies which makes it valuable for this review as all three study types were included in the review. The tool also assesses risk of bias with respect to the included groups, the selection process and the representativeness of the sample (see Ta-ble 3). Based on this assessment we discussed the risk of bias in the discussion session (pages 30 + 33-36). In order to make this clearer, we have added a paragraph explaining the MMAT tool (page 9). 

Reviewer 1. P4: The authors have highlighted using the Kirkpatrick's training evaluation model and created a heading 'quality assessment' page 23 (line 271). I will recommend the heading to be re-phrased along the lines of 'quality assessment of training reported in the included studies' or 'evalu-ation of training in the included studies' (these are merely suggestions). The authors can come-up with a more suitable heading. But traditionally, quality assessment in systematic reviews are usually alluded to assessing the methodological rigor, bias, and reporting of the studies included in the re-view.

Reply: We have rephrased the heading to “Quality assessment of the Programmes”.

Reviewer 1. P5: In Table 2, pages (11-20), it will be useful to have the First author's surname and year of publication in the first column and then add the reference next to it. For example, Andrew, 2010 [18]. This will be more informative for the reader as opposed to the current form.

Reply: This advice is very helpful. We have added the First author’s surname and year of publication in Table 2 and 3. Correspondingly, we have uploaded updated version of supporting information S3 and S4. 

Reviewer 1. P6: A column for study design needs to be added in Table 2 (pages 11-20).

Reply: We have now added a column for study design in Table 2. 

Reviewer 1. P7: The 'quality assessment' results describing the Kirkpatrick's training evaluation model (pages 23-25) will be difficult to comprehend by a lay reader, the authors need to explain to the readers what the %s or ratings mean and their implications. Additionally, the references should be substituted with the First Author's surname and year of publication so the reader knows the score for each included study.

Reply: We can understand this concern very well. Thus, we have added a paragraph explaining the Quality Assessment Tool and the Kirkpatrick’s training evaluation model in which we also explain what the %s and ratings mean (page 9). We have also substituted references with the first Authors surname and year of publi-cation in Table 3. 

Editor: Specialist English review and edit is also recommended 

Reviewer 1. P8: The manuscript should be proof-read once again for punctuation and grammatical issues.

Reply: We apologise for mistakes in the manuscript. The article has now been proof-read by a native speaker. 

Reviewer 2: The introduction notes the number of international migrants, perhaps the number of refugees could also be noted, unless they are included in the migrants category.

Reply: This distinction is truly very important. The number in the introduction refers to migrants as well as refugees as they are included in the category of migrants. 

Reviewer 2: The discussion section on page 29 line 378 includes aspects of limitations. Perhaps these could be moved to pages 30 -31 which discusses the strengths and 

limitations of the study.

Reply: We agree with this concern. Due to the topic of this paper it was important for us, to put these consid-erations at the beginning of the discussion in order to be able to lead the discussion critically. We also wanted the reader to read the discussion in light of the limitations to ensure critical reflection and contextualisation. 

Reviewer 2: There are minor edits that were noted, like the mean age of females in 51 on page 17. 

Reply: We have carefully revised the article and hopefully fixed all the minor mistakes. 

Reviewer 2: On page 22, revisit the abbreviations used to make sure they match content under discus-sion.

Reply: The abbreviations on page 22 refer to the evaluation methods. We did not pick up on these single methods in the discussion session as we focused on the overall quality of the methods. Therefore we may have not understood this point and would kindly request an elaboration on this in order to fix it. 

Editor P.1, 2 and 3: (1) Please ensure that your manuscript meets PLOS ONE's style requirements, in-cluding those for file naming. (2). Please remove your figure 1 from within your manuscript file, leaving only an individual TIFF/EPS image file, uploaded separately as a figure file. This will be au-tomatically included in the reviewers’ PDF. (3). Please ensure that you refer to Figure 1 in your text as, if accepted, production will need this reference to link the reader to the figure.

Reply: We apologise for any deviations from PLOS ONE’s style requirements. We have now revised the man-uscript according to PLOS ONE’s style requirements, removed figure 1 and referred to figure 1 in the text.

All changes throughout the manuscript were highlighted and uploaded along with an unmarked version. We are very grateful for your support that has contributed to the improvement of the article. We hope, that the revised version of the manuscript now meets with your approval. 

We look forward to receiving your reply.

Yours sincerely,

Sidra Khan-Gökkaya

---

## [Editor Report · Decision Letter 1]

25 Oct 2019

Qualification programmes for immigrant health professionals: a systematic review

PONE-D-19-22024R1

Dear Dr. Sidra Khan-Gokkaya,

We are pleased to inform you that your manuscript has been judged scientifically suitable for publication and will be formally accepted for publication once it complies with all outstanding technical requirements.

With kind regards,

Sharon Mary Brownie

Academic Editor

PLOS ONE

Additional Editor Comments (optional):

Reviewers concerns have been addressed
---

## [Editor Report · Acceptance letter]

8 Nov 2019

PONE-D-19-22024R1 

Qualification programmes for immigrant health professionals: a systematic review 

Dear Dr. Khan-Gökkaya:

I am pleased to inform you that your manuscript has been deemed suitable for publication in PLOS ONE. Congratulations! Your manuscript is now with our production department. 

With kind regards,

on behalf of

Professor Sharon Mary Brownie 

Academic Editor

PLOS ONE